# Clinical and Molecular Aspects Associated with Defects in the Transcription Factor POU3F4: A Review

**DOI:** 10.3390/biomedicines11061695

**Published:** 2023-06-12

**Authors:** Emanuele Bernardinelli, Florian Huber, Sebastian Roesch, Silvia Dossena

**Affiliations:** 1Institute of Pharmacology and Toxicology, Paracelsus Medical University, 5020 Salzburg, Austria; e.bernardinelli@pmu.ac.at (E.B.); f.huber@pmu.ac.at (F.H.); 2Department of Otorhinolaryngology, Head and Neck Surgery, Paracelsus Medical University, 5020 Salzburg, Austria

**Keywords:** POU3F4, X-linked deafness, transcription factor, hearing loss, gene variants

## Abstract

X-linked deafness (DFNX) is estimated to account for up to 2% of cases of hereditary hearing loss and occurs in both syndromic and non-syndromic forms. *POU3F4* is the gene most commonly associated with X-linked deafness (DFNX2, DFN3) and accounts for about 50% of the cases of X-linked non-syndromic hearing loss. This gene codes for a transcription factor of the POU family that plays a major role in the development of the middle and inner ear. The clinical features of POU3F4-related hearing loss include a pathognomonic malformation of the inner ear defined as incomplete partition of the cochlea type 3 (IP-III). Often, a perilymphatic gusher is observed upon stapedectomy during surgery, possibly as a consequence of an incomplete separation of the cochlea from the internal auditory canal. Here we present an overview of the pathogenic gene variants of *POU3F4* reported in the literature and discuss the associated clinical features, including hearing loss combined with additional phenotypes such as cognitive and motor developmental delays. Research on the transcriptional targets of POU3F4 in the ear and brain is in its early stages and is expected to greatly advance our understanding of the pathophysiology of POU3F4-linked hearing loss.

## 1. Introduction

Hearing loss is the most common sensory defect and affects up to 5 in 1000 newborns worldwide in a disabling form [1]. The aetiology of hearing loss is multifactorial and includes genetic and environmental factors. Deafness can be the result of physical trauma, infections, or as a side effect of some medications such as diuretics, anti-cancer drugs, and aminoglycoside antibiotics, but it is estimated that genetic causes are responsible for at least 60% of the cases [2]. Multiple genes have been associated with hereditary hearing loss in a syndromic or non-syndromic form. Defects in genes found on the X chromosome are causative of X-linked hearing loss, which accounts for up to 2% of the cases of hereditary deafness and is characterized by a typical inheritance pattern. *POU3F4* is the gene most commonly associated with X-linked hearing loss and accounts for about 50% of these cases [3]. In the present review, we summarize and discuss the genetic variants identified in the *POU3F4* gene that have been associated with hearing loss. We will focus on the type of inner ear malformations commonly associated with defects in *POU3F4* and on the possibility of a syndromic form of X-linked deafness, where defects in organs other than the auditory apparatus are reported in patients with pathogenic variants of *POU3F4*.

## 2. X-Linked Hearing Loss

Most of the genes and loci associated with hearing loss are found on the 22 autosomal chromosomes, but a small percentage of causative genes can also be found on the sex chromosomes. X-linked deafness (DFNX) is estimated to account for up to 2% of the cases of hereditary deafness [3,4]. Due to the pattern of inheritance, hemizygous male individuals are most commonly affected by X-linked deafness, while female heterozygous carriers might display varying degrees of hearing loss. Both syndromic and non-syndromic forms of hearing loss are associated with X-chromosome alterations. Up to date, 17 X-chromosome genes have been associated with a syndromic form of hearing loss and, including *COL4A5*, which causes Alport syndrome, which is characterized by glomerulopathy, hearing loss, and lens abnormalities [5], and *AIFM1*, which is responsible for Charcot-Marie-Tooth syndrome, which is characterized by heterogeneous forms of motor and sensory neuropathies [6]. Some genes, such as *COL4A6*, *PRSP1*, and *AIFM1*, have been identified as causative of both syndromic and non-syndromic X-linked hearing loss [3]. For what concerns X-linked non-syndromic hearing loss, 5 genes have been identified so far: *PRPS1* [7], *SMPX* [8], *AIFM1* [9], *COL4A6* [10], and *POU3F4* [11]. The latter is the gene most commonly responsible for X-linked hearing loss and accounts for about 50% of the cases of X-linked non-syndromic hearing loss [3]. 

## 3. The Transcription Factor POU3F4

The gene coding for the transcription factor POU3F4/BRN4 (OMIM *300039) was identified in 1995 [12] as the gene responsible for X-linked mixed deafness with stapes fixation (DFNX2, DFN3, OMIM #304400). Later, the clinical features of POU3F4-related hearing loss have been better defined by linking defects in the transcription factor with specific inner and middle ear malformations, including an incomplete partition (IP) of the cochlea and thickening of the stapedial footplate, occasionally associated with an enlargement of the vestibular aqueduct (EVA) [13]. Another feature that is often associated with defects in POU3F4 is the occurrence of a cerebrospinal fluid gusher during inner ear surgery [14,15], which will be detailed below.

The transcription factor POU3F4 is expressed in the mesenchymal cells of the neural tube during early embryonic development in mice. It contributes to the embryogenesis and development of the otic capsule into the mature inner and middle ear, including the morphogenesis of the stapedial ossicle [16] and the fasciculation of the spiral ganglion neurons to eventually form the cochlear synapses and the surrounding sensory hair cells [17]. In the mouse model, POU3F4 also promotes the assembly and organization of connexins into plaques, which establish a tight connection between the supporting cells in the cochlea and are essential for an efficient circulation of K^+^ ions and other small metabolites [18]. K^+^ concentration in the endolymph is a crucial factor for the conversion of the mechanical stimulus elicited by sound into the electrical impulse that travels from the hearing organ to the brain. The lack of POU3F4 expression in the mouse model has been shown to result in the degeneration of connexin plaques and defects in the composition of the endolymphatic fluid, with loss of the endocochlear potential [18].

## 4. Clinical and Anatomical Features of POU3F4-Linked Hearing Loss

As it is a form of X-linked hearing loss, males are more often affected by POU3F4-linked hearing loss. Hemizygous males carrying a defective *POU3F4* form typically show a severe degree of sensorineural or mixed hearing loss, while most heterozygous females do not display any significant hearing defect. Nonetheless, a number of cases of female heterozygous carriers displaying hearing loss and vestibular dysfunctions have also been reported [19,20]. 

The type of hearing loss reported in most cases is bilateral, severe to profound, with conductive and sensorineural components. The source of the conductive component is attributed to the defect and fixation of the stapes [21]. ENT surgeons consistently advise against stapedotomy due to the high frequency of severe cerebrospinal fluid (CSF) gusher and resulting loss of cochlear function. Cochlear implant is possible in POU3F4 patients; however, specific complications and adverse events such as CSF gusher and misplacement of the electrode into the internal auditory canal (IAC) need to be anticipated and possibly managed intraoperatively [14,21]. 

The typical morphology of the inner ear in POU3F4 patients has been described as incomplete partition type III (IP-III), first reported by Phelps et al. in 1991 [22] and later classified more concisely by Sennaroglu and Bajin in 2017 [23]. Due to an impaired embryogenic development of the otic capsule in comparison to a healthy cochlea (Figure 1A,B), the modiolus and the base of the cochlea are mostly missing but the interscalar septa are still present and the dimensions of the cochlea are roughly normal (Figure 1C,D). A missing anatomic separation of the cochlea from the IAC is also a typical hallmark of POU3F4-related hearing loss, and it can explain the potential risk of CSF gusher following iatrogenic opening of the cochlea. The vestibular aqueduct is described to show variable degrees of dilatation [21]; however, definite EVA is not found consistently in POU3F4/IP-III patients [24,25]. Still, an EVA has been reported in association with *POU3F4* variants in a number of cases [14,25,26]; therefore, with the present knowledge, it is difficult to establish whether *POU3F4* variants are causative of EVA or a contributing factor. The cochlear nerve can be found intact in all patients, making a cochlear implant supply for POU3F4 patients, albeit challenging, always possible. 

Besides hearing loss, further clinical symptoms are described in individual cases, such as peripheral vestibular dysfunction [27] as well as cognitive and developmental issues, ranging from attention disorders to delayed development, dystaxia, or hemiparalysis [14]. These findings gave rise to a discussion on whether DFNX2 should always be considered non-syndromic hearing loss. However, unfortunately, these symptoms are not consistently reported or systematically investigated, and the information remains sparse.

## 5. Structural Features of the POU Family of Transcription Factors

The transcription factor POU3F4 is classified as a member of the Pit-Oct-Unc (POU) protein family, which takes its name from the transcription factors where this specific DNA-binding domain was first found (mammalian Pit-1, Oct-1, Oct-2, and *C. elegans* Unc86 [28]). All members of the family are characterized by the presence of a bipartite DNA-binding domain consisting of a 76-78-amino-acid-long POU-specific domain (POU_S_) and a C-terminal 60-amino-acid-long homeodomain, which is a homologue of the homeodomain initially identified in *D. melanogaster* (POU_H_) [29,30]. The two domains are separated by a segment with variable length in different members of the family. The bipartite structure grants high structural flexibility, allowing for the binding of DNA in diverse conformations [31]. The three-dimensional structures of both domains have been defined by NMR. The POU_H_ homeodomain structure was first determined in *Drosophila* and showed a typical helix-turn-helix motif interacting with the DNA through a third helix [32]. The POU_S_ domain 3D structure also revealed a compact bundle of four helixes, with the two in the middle organized in a typical helix-turn-helix conformation [33]. The DNA-binding properties of the two domains define the target specificity and the transcriptional activity of the individual members of the family [34]. Several members of the POU family of transcription factors, including POU3F4, have been shown to form homo- and heterodimers with other members of the family or with transcription factors of the Sox family [35]. The homo- and heterodimerization influences target recognition and therefore increases even further the binding flexibility of the transcription factors. 

Six classes of POU transcription factors have been defined according to the specific properties of the POU_S_ domain and the linker segment between POU_S_ and POU_H_ [36]. POU transcription factors play a major role in multiple cell types, both in developmental and homeostatic processes. Several studies on mouse models have shown that some members of the family are ubiquitously expressed (Oct1, Pou2f1 [37]), while others display a spatially and temporally limited expression pattern ranging from the pituitary gland (Pit1, Pou1f1 [38]), hypothalamus, and brain tissue (Brn4, Pou3f4 [39,40]) to the lymphocytes (Oct2, Pou2f2 [41]), skin (Oct11, Pou2f3 [42]), testis (Oct6, Pou3f1 [39]), retina (RPF-1, Pou6f2 [43]), and inner ear (Brn4, Pou3f4 [44]). The role of several members is restricted to the embryonal phase, while others are also expressed and active in the adult phase.

Overexpression of specific POU transcription factors has been shown to act as a tool for cellular reprogramming into other cell types or even pluripotent stem cells [45,46]. Among the POU family of transcription factors, Oct4 (POU5F1) especially has been shown to play a central role in cell fate reprogramming in fibroblasts and other cell types. Cells transfected with a cocktail of transcription factors, including human Oct4, Sox2, Klf4, and c-Myc [47], could be reprogrammed to pluripotency and are commonly defined as induced pluripotent stem cells (iPSCs). The same Brn4 (Pou3f4) was shown to play a similar role in the conversion of mouse fibroblasts, although with a limited degree of reprogramming efficiency when compared to Oct4 [48].

The 8907-bp-long *POU3F4* gene is located on the X chromosome at the Xq21.1 locus and includes one single coding exon spanning 1506 bp (Figure 2A). The encoded transcription factor POU3F4 is a 361-amino-acid protein and possesses the two conserved DNA-binding domains typical of the POU protein family. The POU-specific (POU_S_) domain encompasses the amino acids L194 to D260 (nucleotides from 580 to 780), while the POU-homeodomain (POU_H_) ranges from the amino acid G276 to R335 (nucleotides 826 to 1005) [12]. POU3F4 is predicted to have three nuclear localization signals (NLSs) [25,49] driving its subcellular trafficking, and it has been shown to localize in the nuclear compartment [50]. Two of the predicted NLSs are localized within the POU_H_ domain and include the amino acids from G276 to T283 and from N328 to R335, respectively, while a third one is predicted to be located within the POU_S_ domain and encompasses amino acids F201 to L208 (Figure 2B). Gene variants affecting the NLSs by point mutation, deletion, or truncation are expected to disrupt the trafficking of the protein and therefore result in defective subcellular localization, as shown by us and others [25,49,51]. POU3F4 was first identified in mouse brain tissue and therefore named Brain-4 (Brn4) [40]. The expression of this transcription factor was detected in parts of the brain both during embryonic development and in adult mice. Okazawa et al. showed that Brn4 binds to consensus sequences in the dopamine D1a receptor gene in the striatum in mice [52], suggesting a functional role of Pou3f4/Brn4 in the dopamine pathway. The underlying mechanism and pathophysiological implications of this finding have not been elucidated so far. During embryonic development of the mouse, POU3F4 is highly expressed in the mesenchymal cells surrounding the forming otic vesicle and contributes to the modelling and differentiation of the labyrinthine apparatus [53,54]. The transcription factor is still expressed in the adult mouse inner ear, albeit at much lower levels, and might play a role in the survival and maintenance of spiral ganglion neurons [17].

## 6. Transcriptional Targets of POU3F4

No transcriptional target of POU3F4 has been fully characterized so far. A few potential targets have been identified, but none of them has been fully investigated, especially concerning the POU3F4 binding site in their promoter region [53,55,56], thus leaving a question mark regarding the functional network downstream of this transcription factor. Class III POU transcription factors, including POU3F2, POU3F3, and POU3F4, were found to bind to a non-canonical consensus sequence in an enhancer region driving the expression of the homeobox transcription factor gene *Otx2* in the developing neural tube in mouse models [57]. The binding of POU III transcription factors to the enhancer drives the expression of OTX2 in the developing forebrain and midbrain, whereas the competition of the homeobox gene *Gbx2* in the hindbrain prevents the binding of the POU transcription factors and suppresses the expression of OTX2 in that area.

In the inner ear, POU3F4 has been shown to regulate the expression of the receptor tyrosine kinase EPHA4 [53]. Coate and colleagues demonstrated that the expression of EPHA4 is decreased in *Pou3f4* KO mice and that POU3F4 binds *EphA4* regulatory elements. The lack of EPHA4 expression in the inner ear results in defective fasciculation of the spiral ganglion neurons. The same group showed that *Pou3f4* and the Eph receptor transmembrane ligand *Ephrin-B2* exhibit a common spatio-temporal expression pattern during the organogenesis of the middle and inner ear [56], thus providing evidence for a role for POU3F4 in the development of the bony tissue surrounding the vestibular labyrinth. 

We recently identified a putative transcriptional target of POU3F4 by RNAseq and real-time PCR analysis of cells overexpressing this transcription factor [25]. In these cells, the amino acid transporter SLC6A20 was upregulated by wild-type POU3F4 but not by two of its pathogenic variants. We could also show the co-expression of *Slc6a20* and *Pou3f4* transcripts in the mouse cochlea, paving the way for further investigation of the mechanistic aspects underlying POU3F4 pathophysiology in the inner ear.

POU3F4 homologs and orthologs are also found in lower organisms. Gao and colleagues identified an ortholog of POU3F4 as the transcription factor driving the expression of Aspein and Prismalin-14, two gene products involved in the formation of the oyster shell in the pearl oyster (*Pinctada fucata*) [55]. The promoter regions of the two genes have been investigated, and putative Pf-POU3F4 binding sequences have been identified. No orthologs of the two genes exist in higher organisms; therefore, the targets of POU3F4 in mammalians remain largely unknown.

Mathis et al. identified the sequence 5′ ATGCAAAT 3′ as the octamer binding motif recognized by the transcription factor POU3F4 and other members of the POU family (Figure 2C) [40]. Later, Malik et al. showed that POU3F4 undergoes autoregulation by driving its own transcription. These authors could identify a functional binding motif in the promoter region of the *POU3F4* gene itself [58], setting the basis for the development of a functional test of POU3F4 transcriptional activity.

Thus, albeit the central role of POU3F4 in the development of the middle and inner ear structures is well established, the underlying mechanisms remain vastly unexplored and necessitate further investigations. 

## 7. Variants of POU3F4 Identified in Hearing Loss Patients and Genotype-Phenotype Correlation

Since the definition of *POU3F4* as the causative gene for DFNX2 (DFN3) [12], several gene variants have been identified and reported. *POU3F4* variants include missense, nonsense, and frameshift mutations within the only coding exon of the gene, as well as deletions, insertions, and inversions within the gene or even several kilobases upstream of the gene itself. Large deletions more than 900 kb upstream of the gene are not expected to alter the *POU3F4* coding sequence but possibly affect regulatory regions relevant for its expression [50,59,60]. Deletions of whole portions of the X chromosome, including the coding sequence of *POU3F4*, have also been reported [12] and are often associated with more complex clinical conditions. The ClinVar website of the National Center of Biotechnology Information reports a total of 249 entries involving the *POU3F4* locus (https://www.ncbi.nlm.nih.gov/clinvar/, accessed on 8 March 2023). Many of the variants reported involve large portions of the X chromosome that include *POU3F4*, while 99 variants affect areas smaller than 50 bp in length. Among these variants are deletions, duplications, and insertions involving one or more nucleotides, as well as single nucleotide polymorphisms (SNPs). Regarding their pathogenicity, 174 variants are categorized as pathogenic, 12 as likely pathogenic, 12 as benign, and 17 as likely benign, while 36 are classified as variants of uncertain significance. 

A similar picture can be found on HGMD (https://www.hgmd.cf.ac.uk/ac/search.php, accessed on 8 March 2023), where a total of about 70 variants, divided into missense or nonsense mutations, small deletions, and insertions, are reported. More complex variants involving large portions of the gene or of the X chromosome are also reported. For most of the variants, only limited information regarding their clinical implications is available, and even less information on the molecular and functional aspects of the coded protein product can be found in the literature. Pathogenicity is in most cases determined either by population frequency or pathogenicity prediction algorithms (e.g., SIFT, PolyPhen, and similar), and only in a minority of the studies was a functional or molecular analysis of the variant carried out [25,49,50,51]. The pathogenicity of the few characterized variants was assessed by showing an aberrant subcellular localization, with the mutant protein partially or completely excluded from the nuclear compartment. A partial or complete loss of transcriptional activity was associated with impaired nuclear trafficking. The functional characterization of POU3F4 variants is particularly challenging because no physiological transcriptional target has been unequivocally established. A functional analysis of POU3F4 transcriptional activity could be set up by exploiting the binding motif identified by Mathis et al. or a portion of the *POU3F4* promoter to drive the transcription of a reporter gene upon co-expression with the transcription factor itself [25,49,50].

Table 1 lists the POU3F4 protein variants published up to now in the scientific literature (https://pubmed.ncbi.nlm.nih.gov/, accessed on 8 March 2023) in association with hearing loss. The table reports only single nucleotide variations and small indels. Large structural variants (larger deletions/inversions) and chromosomal rearrangements are not listed. The reference given corresponds to the first report of the variant. Most variants have been identified only once, in individual patients or families. In the same table, the associated clinical features and the pathogenicity assignment according to the ACMG/AMP criteria [61] are also reported. Only a minority of these variants are found in public databases, such as ClinVar (https://www.ncbi.nlm.nih.gov/clinvar/, accessed on 8 March 2023) and the accession number corresponding to each variant is reported when available.

As shown in Table 1, inner ear malformations, and especially a cochlear incomplete partition type III, are reported in most patients carrying POU3F4 variants. In many studies, a malformation of the cochlea was described as displaying the typical features of an incomplete partition type III without explicitly defining it as IP-III [19,83]. This was particularly common before 2017, when Sennaroglu clearly classified the different types of inner ear malformations [23]. While typical features of an IP-III malformation are present in the great majority of patients carrying *POU3F4* pathogenic variants, EVA is reported in few cases. Interestingly, EVA was also reported in a patient carrying a synonymous *POU3F4* nucleotidic conversion, resulting in no change in the amino acid sequence [65]. This *POU3F4* variant is classified as likely non-pathogenic. In the same patient, together with an EVA, a Mondini malformation was also reported. The Mondini malformation, also defined as incomplete partition type II (IP-II) [23], is typically observed in association with an EVA in patients with pathogenic variants in the *SLC26A4* gene encoding the pendrin protein [90], suggesting that the *POU3F4* variant was coincidentally detected in this patient and EVA and that Mondini malformations might be due to other genetic factors. 

In one study, a cochlear malformation described as having similarities with a Mondini dysplasia was reported in two patients in association with two different nonsense *POU3F4* pathogenic variants, both leading to a premature truncation of the protein [51]. The association of this phenotype with pathogenic *POU3F4* variants suggests that a less severe form of IP-III could have been classified as IP-II in these patients.

The penetrance of the enlargement of the vestibular aqueduct seems to be incomplete in POU3F4-related hearing loss, with only a fraction of studies reporting an anomaly in its dimension above the threshold [91]. In the works of Giannantonio et al. [83], Barashkov et al. [19], and Stankovic et al. [14], an enlargement of the semicircular canals was observed in the patients, but no mention of an EVA is explicated. Such malformations might actually hint at a degree of abnormality in the vestibular apparatus, including the vestibular aqueduct.

Vestibular symptoms and/or dystaxia were reported in two unrelated patients, one of which had enlarged semicircular canals [19,27]. In female carriers, the possible presence of vestibular symptoms was verified but not found [20]. In all of the other studies listed in Table 1, vestibular symptoms were not found or had not been assessed.

The great majority of the *POU3F4* variants identified and reported in the literature are found within the sequence coding for one of the two DNA-binding domains, i.e., the POU_S_ domain and the POU_H_ domain (Table 1). Alterations of the domain sequence and structure impair DNA binding and potentially result in loss of transcriptional activity as the underlying mechanism of pathogenicity. Only 28 out of 82 variants are found outside the POU domains, but all of them result in premature termination of the protein, either as a consequence of a nonsense variant or a frameshift. The only exception is the synonymous conversion Val46= reported by Xiao et al. [65]. However, this variant has been classified as likely benign by the authors and was possibly coincidental. The partial or complete loss of the DNA-binding domains remains, therefore, the main pathogenic mechanism underlying POU3F4 defects correlating with the malformative hearing loss phenotype. 

A correlation between the type of protein variant (truncation or amino acid substitution), amino acid position of the truncation, or type of amino acid substitution and the severity of disease (syndromic or non-syndromic) could not be found (Table 1). 

## 8. Hearing Loss Phenotypes Described in Female Carriers

Males are more often affected by POU3F4-related deafness due to its X-linked pattern of transmission, but female carriers have been reported as displaying a wide range of hearing loss phenotypes, with or without inner ear malformations. The hearing defect in female carriers is often mild, without malformations; in some cases, it is only detected upon testing but barely invalidates [20,92]. Nonetheless, a number of cases of severe hearing loss associated with gross malformations of the inner ear have also been reported [20]. The heterogeneity in the phenotypes observed in carrier females is possibly due to unbiased X chromosome inactivation in different cells of the body. If the chromosome carrying the wild-type allele is inactivated, the defective protein coded on the second copy of the X chromosome is expressed. When this happens in the right cells at the right time (e.g., in the developing inner ear during embryogenesis), the hearing loss phenotype can arise. The inactivation of one of the X chromosomes is usually random and can be estimated to range from 50:50 to 75:25 if no other genetic or environmental factors contribute to the distribution of inactivation. Furthermore, the distribution of X chromosome inactivation in different organs and tissues is also expected to be random, making the prediction of the pattern of inactivation within the inner ear difficult. Wu et al. were able to show the rate of randomization of X chromosome inactivation in explanted organs of Corti from genetically engineered mice [93]. From their study, it becomes evident that the high inter- and intra-individual variability in the choice of which X chromosome gets inactivated exists with no recognizable pattern. In the case of the X-linked phenotype, this translates to a random mosaic distribution of cells either producing a fully functional protein or the defective counterpart with a roughly 50:50 frequency. The development of a phenotype is therefore also a stochastic event according to the local distribution of X chromosome inactivation.

One further challenge in female carriers is the detection of the genetic alteration, especially when large genomic deletions are present. In the presence of chromosomal aberrations, the analysis of copy number variations (CNVs) has to be considered in order to detect anomalies in female individuals, who bear two copies of the X chromosome. Such analysis is not always implemented in routine genetic analysis, and Sanger sequencing would identify the wild-type allele and miss the detection of the defective allele. Digital PCR (dPCR) and Multiplex Ligation-dependent Probe Amplification (MLPA) have been successfully employed for the rapid and effective detection of CNVs in families of index patients carrying large genomic deletions and allowed for the detection of heterozygous carrier females [92]. Nonetheless, very few studies focused on the detection of female carriers, and therefore little is known regarding their actual frequency in the general population.

## 9. Further Symptoms Described in POU3F4 Patients

As previously detailed, POU3F4 defects are typically associated with sensorineural or mixed hearing loss with IP-III, and the symptomatic DFNX2 or DFN3 is most often described as a non-syndromic form of hereditary hearing loss. However, in a number of studies, other symptoms in addition to hearing loss have been reported in patients with *POU3F4* variants. Most commonly, mental and physical retardation have been diagnosed [66,83], but more severe cases of motor impairment [14,19] up to partial paralysis [67] have also been reported. Whether these symptoms are directly connected to defects in the transcription factor or whether they are coincidental is still a matter of debate. Some studies point to the existence of a syndromic form of POU3F4-related deafness [83,94]. The incidence of such symptoms is anyway relatively low, either due to a lack of reporting or the actual low incidence of the phenotypes. In the few reported cases, a correlation with a more severe, mixed type of deafness can often be observed [14,19,83]. Whether this is also coincidental can only be matter of speculation.

Very few studies focus explicitly on the cognitive aspects of individuals carrying POU3F4 pathogenic variants. In a recent exploratory study, Smeds et al. specifically investigated multiple aspects of mental development and behavioral traits in a group of young children with IP-III malformations and a cochlear implant (CI) in whom variants in the *POU3F4* gene have been detected [94]. A control group was selected from age- and sex-matched children with CI and no variants in *POU3F4*, but instead carrying pathogenic sequence alterations in the *GJB2* gene, which is the most common genetic cause of non-syndromic deafness in Caucasian populations. From the study, a clear neurodevelopmental deficit in children with IP-III cochleas and *POU3F4* pathogenic variants could be assessed when compared to peers carrying sequence alterations in *GJB2*. The difficulties ranged from poor speech recognition and processing, despite normal measured hearing threshold, to attention deficits and hyperactive behavior. In most of the patients, including the only female, large deletions on the X chromosome were detected, either affecting regulatory regions upstream of *POU3F4* or including the *POU3F4* locus together with other genes potentially related to the development of cognitive abilities [95]. Still, poor speech recognition and processing were observed in all patients, regardless of the type of sequence alterations affecting *POU3F4*. 

Other studies reported poor speech recognition in patients with *POU3F4* pathogenic sequence alterations despite a normal aided hearing threshold [96,97]. The debate is still open on whether cognitive deficits are due to gross chromosomal rearrangements often observed in IP-III patients or are a direct consequence of POU3F4 dysfunction. The transcription factor has been shown to play a central role in spiral ganglion neuron fasciculation and organization through interaction with Epha4 [53]. Therefore, a defect in the neural transmission could explain the sound processing difficulties despite a fully functioning CI. Furthermore, POU3F4 has been shown to be expressed in the brain, hence the initial name of Brn4 [40,54], and although its exact function in the brain is still unknown and no central nervous system-related phenotypes have been shown in animal models [54], an involvement in cognitive development is plausible.

In the present review, we report seven publications describing some form of cognitive or developmental deficit, ranging from attention disorders to intellectual disabilities [14,25,60,67,70,83,88]. The cognitive phenotypes are usually associated with severe to profound mixed hearing loss, while they are less commonly observed in patients with a sensorineural or milder form of hearing loss. Other symptoms have also been reported, in isolated cases, in patients bearing *POU3F4* pathogenic variants. Kanno et al. reported a patient affected by bilateral profound mixed-type hearing loss with an early termination of the protein at amino acid p.102, displaying intellectual disabilities and a form of hemiparalysis [67]. No further detail is provided regarding additional symptoms. Motor dysfunctions have also been reported by Stankovic in a young child affected by severe mixed-type hearing loss [14]. The affected patient carried an early nonsense variant in the CDS of *POU3F4* and displayed an attention disorder and learning retardation associated with a delayed development of motor skills. A similar delay in motor development was also reported by Vore et al. in a child affected by sensorineural hearing loss and carrying a single amino acid conversion within the POU_S_ DNA-binding domain [75]. Barashkov et al. reported a patient with a nonsense variant in *POU3F4* displaying motor impairment and dystaxia [19]. The fact that motor impairments are reported in a number of cases in association with the hearing loss phenotype in patients with IP-III and pathogenic variants in *POU3F4* allows for speculation on the possibility of a syndromic form of hearing loss. Unfortunately, the reports are often limited to a description of the symptoms without further investigation, as is often the case when cognitive phenotypes are reported.

De Kok presented a complex clinical picture in a patient with an amino acid conversion of arginine to serine at position p.330. The patient displayed the typical features associated with POU3F4 pathogenic variants, and symptoms of growth retardation were reported, as well as hyperparathyroidism, subclinical hypothyroidism, and intermittent diarrhea [88]. In addition to POU3F4-related hearing loss, the patient had a familial history of multiple endocrine adenomatosis type 1, which possibly explains the complex symptoms observed.

A unique phenotype was recently reported by Chen et al. [66]. These authors described a 12-year-old patient with a missense variant at the beginning of the POU_H_ DNA-binding domain and within the second NLS who displayed profound hearing loss in association with a cataract. In addition, in this case, no further description of symptoms is provided. Considering its uniqueness, this finding might have been coincidental.

## 10. Conclusions

Pathogenic sequence alterations in *POU3F4* are typically associated with a pathognomonic inner ear malformation described as incomplete partition of the cochlea of type 3 (IP-III). In patients with an IP-III, targeted Sanger sequencing of *POU3F4* will most likely lead to a conclusive genetic diagnosis of hearing loss.

Interestingly, in a number of patients, additional symptoms, often involving growth retardation or a cognitive deficit, are reported. These cases are only sparsely documented, and often no further investigation is carried out to elucidate whether the additional symptoms might be coincidental findings or in fact causally associated with the POU3F4 defect. The possible existence of a syndromic form of POU3F4-related hearing loss is worth further investigation. 

Most of the missense *POU3F4* variants affect one of the two DNA-binding domains, POU_S_ and POU_H_, or the three predicted NLSs. The sequence alterations reported outside the DNA-binding domains result in a premature truncation of the protein, which consequently lacks the whole C-terminal portion, including the DNA-binding domains. Loss of DNA binding and consequent lack of transcriptional activity are therefore the most likely molecular mechanisms underlying POU3F4-linked hearing loss. 

Despite the well-established role of POU3F4 in the aetiology of X-linked hearing loss with IP-III, the mechanistic aspects underlying the clinical phenotype are still poorly understood. The identification and characterization of the transcriptional targets of POU3F4 is an essential step towards a better understanding of the pathomechanism of POU3F4-related hearing loss and the associated phenotypes.

The great majority of POU3F4 pathogenic gene variants lead to premature truncation of the protein products, making a pharmacological rescue of POU3F4 function impracticable. The lack of a precise understanding of the POU3F4 transcriptional targets and their physiological function in the context of hearing loss further limits the identification of possible therapeutic targets. Gene therapy and/or stem cell therapy might represent a potential therapeutic approach for POU3F4-related hearing loss in the future.

## Figures and Tables

**Figure 1 biomedicines-11-01695-f001:**
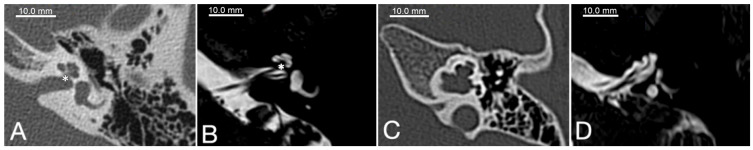
Radiological appearance of a normal cochlea (**A**,**B**) in comparison to a case of incomplete partition type III (**C**,**D**). All images represent axial planes of a left temporal bone. (**A**) and (**C**) are CT-scans; (**B**) and (**D**) are MRI T2-weighed axial scans. The cochlear base (*) can be clearly identified in (**A**) and (**B**) (normal cochlea), whereas it is missing in (**C**) and (**D**) (IP-III cochlea).

**Figure 2 biomedicines-11-01695-f002:**
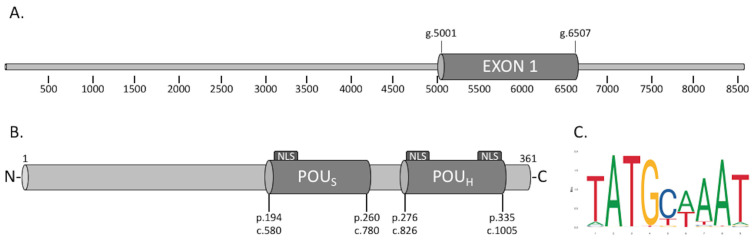
Graphic representation of *POU3F4* gene and protein structure. (**A**) The 8500-bp-long *POU3F4* gene (NG_009936.2) includes one single coding exon spanning g.5001 to g.6507. The coding sequence includes the nucleotides from c.5065 to c.6150. (**B**) Schematic representation of POU3F4 protein sequence. NLS, nuclear localization sequence; POU_H_, POU homeodomain DNA-binding domain; POU_S_, POU-specific DNA-binding domain. The numbers below the graphic indicate the starting and ending points of the bipartite DNA-binding domain. p. indicates the amino acid position (NP_000298.3), c. the nucleotide position (NM_000307.5). (**C**) Predicted POU3F4 DNA-binding consensus sequence (https://jaspar.genereg.net/matrix/MA0789.1/, accessed on 23 May 2023).

**Table 1 biomedicines-11-01695-t001:** Protein variants of POU3F4 reported in the literature and associated clinical features. Amino acid positions are based on NP_000298.3, nucleotide positions are based on NM_000307.5. The sequencing method employed in the individual studies is reported. NGS Panel, next generation sequencing of a panel of hearing loss-related genes, WES, whole exome sequencing, PCR/SSC, PCR Single Strand Conformation assay. The domain affected by the individual variants is listed with an asterisk (*) when the nucleotide change results in a premature truncation of the protein products. In the description of the inner ear malformations, # indicates that an IP-III was not identified as such by the authors of the study, but their description of the cochlear malformation corresponded to the typical features of an IP-III, according to [23]. The pathogenicity was assigned by the authors of the present review, or by the authors of the individual studies when reported in italics, according to the ACMG/AMP criteria [61]. P, pathogenic, LP, likely pathogenic, LB, likely benign, NA, not assigned. Superscripts next to the reference indicate the type of study. C, cohort study, S, single report, where the study focused on few individuals or families affected by hearing loss.

Sequence Information				Clinical Information						Pathogenicity	Ref.
Amino Acid Change	Nucleotide Change	Sequencing Technique	Domain	Ethnicity	Hearing Loss		Malformations	Other Findings	ClinVar		
					Severity	Type					
Ser22Cysfs*19	66del	NGS Panel	N-term*	Chinese	NA	NA	NA	-	1185600	*P* PVS1+PS2+PM2	[62] ^C^
Gln26∗	76C > T	WES	N-term*	Chinese	Severe	NA	Hypoplastic inferior gyrus of cochlea, absence of spiral plate of inferior gyrus	-		P PVS1+PP1+PP4	[63] ^S^
Gln27*	79C > T	WES	N-term*	NA	Profound	SNHL	IP-III	-		*P* PVS1+PP1+PP4	[64] ^C^
Val46=	138T > C	Sanger	N-term	Chinese	Profound	SNHL	EVA, Mondini	-		*LB* BP7	[65] ^S^
Trp57*	171G > A	Sanger	N-term*	Chinese	Profound	NA	IP-III	Developmental retardation		P PVS1+PS2+PP4	[66] ^C^
Ser74Alafs*8	c.220delA	Sanger	N-term*	NA	Severe	Mixed	EVA, IP-III	-	1701827	*P* PVS1+PS3+PP1	[25] ^S^
Gln79*	235C > T	NGS Panel	N-term*	Israeli	Profound	SNHL	IP-II (Mondini)	-	236062	*LP* PVS1+PS3	[51] ^S^
Arg85Alafs*17	249delC	Sanger	N-term*	Japanese	Profound	Mixed	EVA, IP-III	Mental issues, hemiparalysis		*LP* PVS1+PP1+PP4	[67] ^C^
Ser98*	NA	Sanger	N-term*	NA	Mild	Mixed	Bilateral dilatation of internal auditory canals	-		LP PVS1+PP1	[20] ^C^
Trp114*	341G > A	Sanger	N-term*	Pakistani	Severe to profound	Mixed	IP-III#	-	43346	P PVS1+PM2+PP1+PP4	[68] ^S^
Ala116fs*	346delG	Sanger	N-term*	Korean	Severe	NA	IP-III#	-		P PVS1+PP1+PP4	[50] ^S^
Ala116Glyfs*77	346dup	NGS Panel	N-term*	Chinese	NA	NA	IP-III	-		*LP* PVS1+PM2+PP4	[69] ^C^
Ser117Argfs*26	346_350dup	NGS Panel	N-term*	Chinese	NA	SNHL	IP-III	-		*LP* PVS1+PM2+PP4	[69] ^C^
Gly128fs*	383delG	Sanger	N-term*	Korean	Profound	NA	IP-III#	-		P PVS1+PP1+PP4	[70] ^S^
Gln136*	406C > T	Sanger	N-term*	Pakistani	Severe to profound	Mixed	Mid and apical turns not clearly partitioned wide modiolus	-		*P* PVS1+PM2+PP1+PP4	[68] ^S^
Gln136Leufs*58	400_401insACTC	WES	N-term*	Chinese	Profound	SNHL	IP-III	-		*P* PVS1+PP1+PP4	[71] ^S^
Gln136Leufs*58	401_404dup	NGS Panel	N-term*	Chinese	NA	SNHL	IP-III	-		*LP* PVS1+PM2+PP4	[69] ^C^
Val141*	421_422delinsTA	NGS Panel	N-term*	Chinese	NA	Mixed	IP-III	-		*LP* PVS1+PM2+PP4	[69] ^C^
His147Glnfs*94	441del	NGS Panel	N-term*	Chinese	NA	SNHL	IP-III	-		*LP* PVS1+PM2+PP4	[69] ^C^
Pro153Leufs*88	c.458delC	WES	N-term*	Korean	Profound	Mixed	IP-III	-		*LP* *PVS1+PM2+PP3+PP4*	[72] ^C^
Pro164Argfs*77	491delC	Sanger	N-term*	Japanese	Moderate	Mixed	EVA, IP-III	-		*LP* PVS1+PP1+PP4	[67] ^C^
Arg167*	499C > T	Sanger	N-term*	NA	Severe to profound	Mixed	IP-III#, dysplasia of vestibule and semicircular canals	Attention disorders, learning delays, motor delays	43347	*LP* PVS1+PP1+PP4	[14] ^S^
Ser177*	530C > A	Sanger	N-term*	Chinese	Profound	SNHL	IP-III, vestibule dysplasia	-	988337	LP PVS1+PP1+PP4	[26] ^S^
Gln181*	541C > T	Sanger	N-term*	Chinese	Severe	NA	IP-III	-		LP PVS1+PP1+PP4	[66] ^C^
Glu187*	559G > T	WES	N-term*	Caucasian	Severe	SNHL	IP-III	-		*P* PVS1+PM2+PP4	[64] ^C^
Phe201/Lys202del	601–606del TTCAAA	Sanger	POU_S_ NLS	Japanese	Severe	Mixed	IP-III#	-		LP PP1+PM1+PP4	[73] ^S^
Lys202Glu	604A > G	WES	POU_S_ NLS	Vietnam	NA	NA	NA	-		*VUS* PM1+PM2	[74] ^C^
Gln203fs*	607_610delCAAA	Sanger	POU_S_* NLS	NA	NA	SNHL	NA	Vestibular dysfunction	228391	*P* PVS1+PM1	[27] ^C^
Arg204Lysfs*21	610del	NGS Panel	POU_S_ NLS	NA	Severe	NA	NA	-	1185660	*P* PVS1+PM1+PM2	[62] ^C^
Arg205del	614_616delGAA	Sanger	POU_S_ NLS	Chinese	Severe	NA	IP-III	-		LP PM1+PM2+PP4	[66] ^C^
Leu208*	623T > A	Sanger	POU_S_* NLS	Chinese	Profound		IP-III#	Mild mental retardation		P PVS1+PP1+PP4	[70] ^S^
Ser228Leu	NA	Sanger	POU_S_	Korean	Profound	SNHL	IP-III#	Delayed motor development		LP PM1+PP1+PP4	[75] ^S^
Gln229Arg	c.626A>G	WES	POU_S_	NA	Profound	Mixed	IP-III	-		LP PM1+PP1+PP4	[72] ^C^
Thr211Met	632C > T	Sanger	POU_S_	Korean	Moderate	NA	IP-III	-		*LP* PM3+PS3+PP4	[49] ^S^
Val215Gly	644T > G	Sanger	POU_S_	Chinese	Severe	NA	IP-III	-		*LP* *PM1+PM2+PP1+PP4*	[66] ^C^
Gly216Glu	647G > A	Sanger	POU_S_	Chinese	Profound (males)/mild to moderate (females)	Mixed (males)/SNHL (females)	IP-III#	-	422492	LP PM1+PM2+PP1+PP4	[76] ^S^
Leu217Valfs*9	648dupG	Sanger	POU_S_*	Chinese	Profound	NA	IP-III	-		P PVS1+PM1+PP4	[66] ^C^
Leu217*	650T > A	WES	POU_S_*	Caucasian	Profound	SNHL	IP-III	-		*P* PVS1+PM1+PM2+PP4	[64] ^C^
Tyr223*	669 T > A	Sanger	POU_S_*	Chinese	Profound	SNHL	IP-III	-	1185618	P PVS1+PM1+PM2+PP4	[77] ^C^
Gln229Arg	686A > G	Sanger	POU_S_	Korean	Profound	NA	IP-III	-	1879006	*LP* PM1+PS3+PP1+PP4	[49] ^S^
Thr231Ile	692C > T	NGS Panel	POU_S_	Caucasian	NA	SNHL	NA	-		*LP* *PM1+PM2+PP3*	[78] ^C^
Cys233*	699C > A	WES	POU_S_*	Chinese	Severe to Profound	SNHL	IP-III	-		P PVS1+PM1+PP1+PP4	[79] ^C^
Glu236Asp	NA	Sanger	POU_S_	NA	Normal	NA	Normal	-		NA PM1+PP1	[20] ^C^
Glu236Lys	706G > A	NGS Panel	POU_S_	Pakistani	Profound	NA	-	-		*LP* PM1+PM2+PP1	[80] ^S^
Glu236Ala	707A > C	Sanger	POU_S_	Mixed	Profound	SNHL	IP-III#	-		*P* PM1+PM2+PP1+PP4	[81] ^S^
Asn244Lysfs*26	727_728insA	NGS Panel	POU_S_*	Japanese	Severe	SNHL	IP-III	-		P PVS1+PM1+PP4	[82] ^C^
Glu258Argfs*30	772delG	Sanger	POU_S_*	Mixed	Moderate to severe	SNHL	IP-III	-		P PVS1+PM1+PM2+PP1+PP4	[81] ^S^
Gln275*	823C > T	WES	Linker*	Caucasian	Severe	SNHL	IP-III	-		*P* PVS1+PM1+PP1+PP4	[64] ^C^
Arg282Gln	845G > A	Sanger	POU_H_ NLS	Chinese	Profound	NA	IP-III	Cataract		*LP* *PM1+PM2+PP1+PP4*	[66] ^C^
Ile285Asn	NA	Sanger	POU_H_	NA	Normal	NA	NA	-		NA PM1+PP1	[20] ^C^
Ile285Argfs*43	853_854del	NGS Panel	POU_H_*	Israeli	Profound (male)/mild (female)	NA	IP-II (Mondini)	-	43352	*P* PVS1+PM1+PP1	[51] ^S^
Ser288Cysfs*40	NA	Sanger	POU_H_*	NA	Normal	NA	NA	-		NA PVS1+PM1+PP1	[20] ^C^
Val289fs	862del4	Sanger	POU_H_*	Caucasian	Profound	Mixed	NA	-	11681	P PVS1+PM1+PP1	[60] ^S^
Lys290Asn	870G > T	WES	POU_H_	Caucasian	Severe	Mixed	IP-III#, enlarged semicircular canals	Concentration difficulties, developmental retardation		*LP* *PM1+PM2+PP1+PP4*	[83] ^S^
Leu293Val	877C > G	NGS Panel	POU_H_	Chinese	Severe	NA	NA	-	417622	*LP* PM1+PM2	[62] ^C^
Glu294Gly	881A > G	NGS Panel	POU_H_	Chinese	NA	NA	NA	-		*LP* *PM1+PM2+PP1+PP3*	[84] ^C^
Pro301Leu	902C > T	Sanger	POU_H_	Mixed	Severe	Mixed	NA	-		*P* PM1+PM2+PP1	[81] ^S^
Lys302Alafs*25	903_912 delins TGCCA	Sanger	POU_H_*	Chinese	Moderate	NA	IP-III	-		P PVS1+PM1+PP1+PP4	[66] ^C^
Gln306*	916C > T	WES	POU_H_	Caucasian	Severe	SNHL	IP-III	-	1799532	*P* PVS1+PM1+PM2+PP4	[64] ^C^
Ile308Asn	NA	Sanger	POU_H_	NA	Moderate	Mixed	Normal	-		NA PM1+PP1	[20] ^C^
Ile308Ilefs*28	NA	Sanger	POU_H_*	NA	Normal	-	Normal	-		NA PVS1+PM1+PP1	[20] ^C^
Ser309Pro	925T > C	Sanger	POU_H_	Chinese	Profound	Mixed	IP-III#	Mild sinus arrhythmia		LP PM1+PM2+PP4	[85] ^S^
Ser310del	927-929delCTC	Sanger	POU_H_	Korean	Severe	Mixed	IP-III	-	43353	*P* PM1+PS3+PP1+PP4	[50] ^S^
Ala312Pro	934G > C	NGS Panel	POU_H_	Caucasian	Profound	Mixed	IP-III	-		*LP* PM1+PM2+PP1+PP4	[86] ^S^
Ala312Val	935C > T	Sanger	POU_H_	Caucasian	Profound	SNHL	IP-III#	Learning difficulties	11682	LP PM1+PP1+PP4	[60] ^S^
Gln316*	946C > T	Sanger	POU_H_*	Chinese	Severe	NA	IP-III	-		P PVS1+PM1+PM2+PP1+PP4	[66] ^C^
Leu317*	950 T > A	NGS Panel	POU_H_*	Chinese	NA	NA	NA	-		P PVS1+PM1+PM2	[87] ^C^
Leu317Phefs*12	950dupT	Sanger	POU_H_*	Korean	Severe	NA	IP-III	-	41457	*LP* PVS1+PM1+PM2+PS3+PP1+PP4	[49] ^S^
Glu320*	c.958G>T	WES	POU_H_*	Chinese	Moderate to severe	Mixed	IP-III	-		*LP* *PVS1+PM2+PP4*	[72] ^C^
Val321Gly	962T > G	WES	POU_H_	Chinese	Severe	SNHL	IP-III	-		*LP* *PS1+PM1+PM2+PP1+* *PP4*	[66,69] ^C^
Arg323Gly	967C > G	PCR/SSC	POU_H_	Caucasian	NA	Mixed	dilation of semicircular canals and IAM	-	11684	*P* PM1+PP4	[88] ^S^
Val324Asp	971T > A	WES	POU_H_	Caucasian	Severe	Mixed	IP-III	-		*P* PM1+PM2+PP1+PP4	[64] ^C^
Trp325Glyfs*12	973delT	Sanger	POU_H_*	Chinese	NA	Mixed	IP-III	-		*P* PVS1+PM1+PP1+PP4	[77] ^C^
Trp325Arg	973 T > A	Sanger	POU_H_	Caucasian	Profound	SNHL	dysplastic cochlea on both sides, flat angle of the facial nerve	-		LP PP1+PM1+PP3	[89] ^S^
Trp325*	975G > A	Sanger	POU_H_*	Siberia	NA	Mixed	IP-III#, enlarged semicircular canals	Vestibular dysfunction, dystaxia	599350	*P* PVS1+PM1+PM2+PP1+PP4	[19] ^S^
Cys327*	c.979T > A	Sanger	POU_H_*	Caucasian	Profound to Severe	Mixed	EVA, IP-III	Cognitive deficit	1701826	*P* PVS1+PM1+PS3+PP1+ PP4	[25] ^S^
Arg329Gly	985C > G	NGS Panel	POU_H_ NLS	Chinese	NA	NA	NA	-	1297060	*LP* PM1+PM2	[62] ^C^
Arg329Pro	986 G > C	Sanger	POU_H_ NLS	Korean	Severe	Mixed	IP-III	-		P PM1+PS3+PP4+PP1	[50] ^S^
Ile308Thr	987T > C	Sanger	POU_H_ NLS	Mixed	Severe	Mixed	NA	-		*P* PM1+PM2+PP1	[81] ^S^
Arg330Lys	c.989G>A	WES	POU_H_ NLS	Chinese	Severe	Mixed	IP-III	-		*LP* *PS2+PM2+PP3+PP4*	[72] ^C^
Arg330Ser	990A > T	PCR/SSC	POU_H_ NLS	Caucasian	NA	NA	Enlarged semicircular canals IAM	Growth retardation, subclinical hypothyroidism, diarrhea	11683	*P* PM1	[88] ^S^
Gln331Pro	992A > C	Sanger	POU_H_ NLS	Japan	Left: normal/Right: moderate	Mixed	EVA, IP-III	-		LP PM1+PM2+PP1+PP4	[67] ^C^
Thr354Glnfs*115	1069delA	Sanger	C-term ext.	Korean	Profound	NA	IP-III	-		*LP* PM2+PS3+PP1+PP4	[49] ^S^
*362Argext*113	1084T > C	Sanger	C-term ext.	Korean	Profound	NA	IP-III	-		*LP* PM2+PS3+PP4	[49] ^S^

## Data Availability

No new data were created or analyzed in this study. Data sharing is not applicable to this article.

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
