# Peer review of "Clinical and Molecular Aspects Associated with Defects in the Transcription Factor POU3F4: A Review"

_biomedicines, 2023, doi:10.3390/biomedicines11061695_

Round 1

Reviewer 1 Report

This is a review of the POU3F4 transcription factor, a gene causing X-linked hearing loss.

The work is of interest, but it also lacks a structured methodology. The authors should re-write the methodology including a search strategy, period of the study, assess the quality of the studies and perform a synthesis. The ethnic origin of cases is not reported and probably, this is a relevant factor to investigate if there is a founder effect.

The inclusion of all reported cases without a minimum quality assessment gives the same weight to single case report and case series and the authors should consider this issue.

Major considerations

Table 1 is a list of coding variants ranked by the position in the protein. The number of individuals carrier of each variant is missing and the allelic frequency reported on each population should be added for comparison.

The sequencing technology used on each study is not detailed (Sanger, exome, genome) and may explain discrepancies between studies.

Probably it is more logical to split the table in frameshift and missense variants, since frameshift variants are likely to be pathogenic.

Given that many individuals have a IP-III or EVA, it would be interesting to assess vestibular impairment in these cases and try to establish a correlation between each mutation and the syndromic/ non syndromic phenotype.

Conclusions are not well organized, and it is more and extension of the discussion. 2-3 sentences should be enough to summarize it.

I consider that this review should be improved in the methods, the analysis of the data and refine the conclusions.

Author Response

Reviewer 1

This is a review of the POU3F4 transcription factor, a gene causing X-linked hearing loss.

The work is of interest, but it also lacks a structured methodology. The authors should re-write the methodology including a search strategy, period of the study, assess the quality of the studies and perform a synthesis. The ethnic origin of cases is not reported and probably, this is a relevant factor to investigate if there is a founder effect.

The inclusion of all reported cases without a minimum quality assessment gives the same weight to single case report and case series and the authors should consider this issue.

Answer: we are pleased that this reviewer found our manuscript to be of interest and would like to thank her/him for her/his helpful comments. However, we would like to point out that the suggestions given by this reviewer apply to systematic reviews. The aim of the present manuscript is to provide an overview of the published information on POU3F4 pathogenic variants associated with hearing loss in an attempt to identify a correlation between clinical and molecular features. The authors feel that a comprehensive review, rather than a systematic review, is more suited to this aim. The search strategy of data included in Table 1 is now better described at pag. 6 as follows: “Table 1 lists the POU3F4 gene variants published up to now in the scientific literature (https://pubmed.ncbi.nlm.nih.gov/) in association with hearing loss. (....) The reference given refers to the first report of the gene variant”. Therefore, all studies listed in Table 1 have been extracted from the NCBI Pubmed database and include both single reports as well as studies involving larger cohorts of patients from different geographic areas. As suggested by the reviewer, the ethnic origin of patients is now reported in Table 1, when available. In the opinion of the authors, single case reports and case series have indeed the same weight concerning identification of novel gene variants. As an assessment of the quality of the study, the type of study (C: cohort study, S: single report) and the sequencing technology is now reported in a new table (Table 2). In the opinion of the authors, further evaluation of the quality of the studies as well as a systematic review of the published data is outside the scope of the manuscript.

Major considerations

Table 1 is a list of coding variants ranked by the position in the protein. The number of individuals carrier of each variant is missing and the allelic frequency reported on each population should be added for comparison.

The sequencing technology used on each study is not detailed (Sanger, exome, genome) and may explain discrepancies between studies.

Probably it is more logical to split the table in frameshift and missense variants, since frameshift variants are likely to be pathogenic.

Given that many individuals have a IP-III or EVA, it would be interesting to assess vestibular impairment in these cases and try to establish a correlation between each mutation and the syndromic/ non syndromic phenotype.

Answer: We would like to thank the reviewer for these helpful suggestions. We would like to point out that the great majority of variants included in Table 1 were found in only one patient or family, and large population studies are often not available. This is now specified in the text, as follows (pag. 6): “The reference given corresponds to the first report of the variant and is often the only report. Most variants have been identified in individual patients or families”. Nevertheless, a new table (Table 2) including the minor allele frequency (MAF) of individual variants in the reference population and the corresponding reference ID from the NCBI SNP database and/or the ClinVar database, if available, has been added to the manuscript. Also, information regarding the sequencing method has been included in this new table. After thorough discussion, and for ease of consultation, the authors preferred to list the coding variants ranked by the position in the protein.

The possible presence of vestibular symptoms has been verified in only a minority of studies. A short paragraph commenting on this was added at pag. 9.

A sentence concerning the lack of genotype-phenotype correlation was added at pag. 9.

Conclusions are not well organized, and it is more and extension of the discussion. 2-3 sentences should be enough to summarize it.

I consider that this review should be improved in the methods, the analysis of the data and refine the conclusions.

Answer: Thank you for these constructive criticisms. The conclusion section has been revised also to implement the suggestions of other reviewers. Five clear statements are given to draw conclusions, identify lack of knowledge, and suggest future perspectives.

Reviewer 2 Report

The manuscript from Bernardinelli, et al. “Clinical and molecular aspects associated with defects in the transcription factor POU3F4: A review” summarizes the current knowledge on POU3F4 disease-causing variants, and its function in x-linked hearing loss. In my opinion, the review manuscript is overall well written, follows a comprehensive logic, is concise on its main topic and presents several essential studies to this interesting topic. Although, I must raise a few points of criticism:

Major points: 

 -reference [82]/table 1: The refence to Lin et al. describes a novel POU4F3 mutation (resulting in Lys328Glu), but not a POU3F4 variant! The original Scientific Reports publication also displays erroneous gene naming in its abstract in in its main text. Please carefully recheck the original publication, if it is relevant for your review on POU3F4 and change/delete given information in table 1 linked to this publication. The authors might even want to add a short comment on other POU-transcription factors linked to human diseases or hearing loss (such as POU4F3/ BRN3C variants result in autosomal dominant HL) and to highlight differences between these two rather similar named genes.  

- line 54, line 106 ff., and other positions in the manuscript: Please check for correct gene/protein name nomenclature in different species. Human gene names: POU3F4 (italicized, with all letters in uppercase); human and mouse protein name: POU3F4 (not italicized, with all letters in uppercase); mouse gene names: Pou3f4 (often italicized, only first letter in uppercase).

Refer to current species nomenclature databases, like https://www.genenames.org/about/guidelines/

- line 61, ff.,  line 150, ff., and partly at other positions in the manuscript: The presented functional and expression studies in these chapters are describing POU3F4 function/POU3F4 expression in different animal models and intermingle these with the discussed expression patterns/functions in humans (mostly described in the upstream chapters or along the functional studies in other species). Please clarify these species-differences by adding comments/species indications to the sentences of the chapter (e.g.: …during early development in mice.”) and recheck for correct gene nomenclature in different species, like mouse/rodents. In my opinion POU3F4 functions and expression domains during development can be evolutionary conserved in mammals but cannot easily be generalized to humans without further experimental proof. Therefore, precise indication of investigated species/model system is essential to contextualize the presented experimental findings for POU3F4 functions and expression.     

Minor points: 

- line 18: I would suggest to precise the sentence to “.. pathogenic genetic variants of POU3F4…”

- line 51: The authors might want to add references for described x-linked deafness genes PRPS1, SMPX, AIFM1 COL4A6 and POU3F4. A comma missing between AIFM and Col4A6.

- fig. 1: VA and asterisk marks on pictures B and C are hard to see, as they are very small. Please enhance contrast of these marks (e.g. by colors) or enlarge these. Suggestion to authors: To emphasize the patient phenotype you can also mark the expected position of the missing cochlear base in A.

- line 111: “… homologous of the …” the adjective should be changed into “homologue” (noun).

- line 116: “… with the two in the middle assuming a typical helix-116 turn-helix conformation.” To assume should be changed to a better fitting verb, e.g. resembling, building, organizing into.

- line 189: The text at this position could be more precise, e.g. “…, several disease-causing gene variants have been identified…”.

- line 234: The text states “loss of DNA binding domains”, which would imply a complete deletion of these after mutation, or do the authors rather mean “improper function of these domains result in lack of protein ability to bind DNA”? Please specify the text at this position.

Author Response

Reviewer 2

The manuscript from Bernardinelli, et al. “Clinical and molecular aspects associated with defects in the transcription factor POU3F4: A review” summarizes the current knowledge on POU3F4 disease-causing variants, and its function in x-linked hearing loss. In my opinion, the review manuscript is overall well written, follows a comprehensive logic, is concise on its main topic and presents several essential studies to this interesting topic. Although, I must raise a few points of criticism:

Answer: the authors would like to thank this reviewer for the positive comments and the constructive criticisms.

Major points: 

 -reference [82]/table 1: The refence to Lin et al. describes a novel POU4F3 mutation (resulting in Lys328Glu), but not a POU3F4 variant! The original Scientific Reports publication also displays erroneous gene naming in its abstract in in its main text. Please carefully recheck the original publication, if it is relevant for your review on POU3F4 and change/delete given information in table 1 linked to this publication. The authors might even want to add a short comment on other POU-transcription factors linked to human diseases or hearing loss (such as POU4F3/ BRN3C variants result in autosomal dominant HL) and to highlight differences between these two rather similar named genes.  

Answer: The authors sincerely thank the referee for pointing out the mistake. The entry in Table 1 has been removed, as well as the corresponding reference.  

- line 54, line 106 ff., and other positions in the manuscript: Please check for correct gene/protein name nomenclature in different species. Human gene names: POU3F4 (italicized, with all letters in uppercase); human and mouse protein name: POU3F4 (not italicized, with all letters in uppercase); mouse gene names: Pou3f4 (often italicized, only first letter in uppercase).

Refer to current species nomenclature databases, like https://www.genenames.org/about/guidelines/

Answer: Thank you for the observation. The whole text of the manuscript has been thoroughly revised and nomenclature has been corrected according to the current guidelines.

- line 61, ff.,  line 150, ff., and partly at other positions in the manuscript: The presented functional and expression studies in these chapters are describing POU3F4 function/POU3F4 expression in different animal models and intermingle these with the discussed expression patterns/functions in humans (mostly described in the upstream chapters or along the functional studies in other species). Please clarify these species-differences by adding comments/species indications to the sentences of the chapter (e.g.: …during early development in mice.”) and recheck for correct gene nomenclature in different species, like mouse/rodents. In my opinion POU3F4 functions and expression domains during development can be evolutionary conserved in mammals but cannot easily be generalized to humans without further experimental proof. Therefore, precise indication of investigated species/model system is essential to contextualize the presented experimental findings for POU3F4 functions and expression.     

Answer: More precise indications of the species have been implemented in the text where necessary. We do understand there might have been confusion regarding the human/mouse distinction in some paragraphs. We think the issue has been properly fixed.

Minor points: 

- line 18: I would suggest to precise the sentence to “.. pathogenic genetic variants of POU3F4…”

- line 51: The authors might want to add references for described x-linked deafness genes PRPS1, SMPX, AIFM1 COL4A6 and POU3F4. A comma missing between AIFM and Col4A6.

- fig. 1: VA and asterisk marks on pictures B and C are hard to see, as they are very small. Please enhance contrast of these marks (e.g. by colors) or enlarge these. Suggestion to authors: To emphasize the patient phenotype you can also mark the expected position of the missing cochlear base in A.

- line 111: “… homologous of the …” the adjective should be changed into “homologue” (noun).

- line 116: “… with the two in the middle assuming a typical helix-116 turn-helix conformation.” To assume should be changed to a better fitting verb, e.g. resembling, building, organizing into.

- line 189: The text at this position could be more precise, e.g. “…, several disease-causing gene variants have been identified…”.

- line 234: The text states “loss of DNA binding domains”, which would imply a complete deletion of these after mutation, or do the authors rather mean “improper function of these domains result in lack of protein ability to bind DNA”? Please specify the text at this position.

Answer: All of the suggestions have been implemented in the revised text. Individual references have been added for each of the listed X-linked deafness genes. As suggested by the reviewer, Figure 1 has been completely revised and the described anatomical features have been better highlighted.

Reviewer 3 Report

The review from Bernardinelli and colleagues presents summary of knowledge on the role of the POU3F4 gene in X-linked hearing loss. While the topic discussed is of interest for other researchers in the field, overall the manuscript would benefit from extensive revisions. 

Specifically:
1. When discussing X-linked hearing loss, a mention to how many genes associated with syndromic forms of X-linked deafness may be added, together with the information that some genes (such as PRPS1) are linked to both syndromic and non-syndromic forms. 

2.The logical flow of the manuscript can be improved. In few points the authors seem to move forward on a different subject but then come back to the previous one (for example, in section 4 the mention to additional clinical symptoms -which are present in a minority of patients- may probably be moved after the description of the ear morphology in POU3F4 patients). I would also suggest verifying repetition of the same concepts in different sections.

3. Given that this is a review article, I would suggest making better use of the illustrations to convey the most important information also in a visual way. Currently, only two Figures are included, and they are not particularly clear or informative.
Figure 1 is difficult to interpret if you are not an expert, it lacks clear labelling and any reference to the size/dimension. Are all the three images at the same magnification? If possible, adding a drawing of the cochlear structure would help the interpretation of the radiological images. The legend to the figure could be improved. The order of the panels (A, B, C) should follow the same order in which the panels are mentioned in the text (currently, panel 1C is mentioned before panel 1B).
Also, Figure 2 can be implemented to convey more information. For instance, the authors may add a panel with the gene structure, and/or a panel with the consensus sequence logo of the Pou3f4 DNA binding site (publicly available in websites such as the JASPAR CORE database).

4. Among possible targets of Pou3f4, the authors do not mention Otx2 (Inoue F, Kurokawa D, Takahashi M, Aizawa S. Gbx2 directly restricts Otx2 expression to forebrain and midbrain, competing with class III POU factors. Mol Cell Biol. 2012 Jul;32(13):2618-27. doi: 10.1128/MCB.00083-12), which has been demonstrated to be regulated by POU transcription factors, including Pou3f4, by binding a non-canonical target sequence.
Also, there is no mention to the fact that Pou3f4, as other POU transcription factors, can act as a monomer or a dimer (homodimer or heterodimers), thus making even more complicated to define precisely its regulatory network.

5. When presenting POU3F4 variants that have been reported in the literature (Table 1), 
it would be beneficial if the authors will evaluate the pathogenicity of the variants with some dedicated guidelines, for example using ACMG criteria, instead than reporting the pathogenicity assignment from the original work. In addition, it could be useful to report whether the mutations hit a specific functional domain/motif of the protein (e.g. POUs, POUh, NLS, linker). Finally, please specify that the table reports only SNVs and small indels, not structural variants (larger deletions/inversions, etc.).

6. Is it possible that some of the reported pathogenic variant will cause a gain of function of the transcription factor (for example altering the specificity of DNA binding or the specific interaction with protein partners)?

7. The term “mental retardation” would be better substituted with “intellectual disability”, with an indication of the severity -if available.

There are some grammar errors and typos throughout the manuscript. The manuscript would benefit from proof reading.

Author Response

Reviewer 3

The review from Bernardinelli and colleagues presents summary of knowledge on the role of the POU3F4 gene in X-linked hearing loss. While the topic discussed is of interest for other researchers in the field, overall the manuscript would benefit from extensive revisions. 

Answer: we are pleased that this reviewer found our manuscript to be of interest and would like to thank her/him for the constructive criticisms.

Specifically:
1. When discussing X-linked hearing loss, a mention to how many genes associated with syndromic forms of X-linked deafness may be added, together with the information that some genes (such as PRPS1) are linked to both syndromic and non-syndromic forms. 

Answer: More precise information on the genes associated with X-linked syndromic hearing loss has been implemented in the revised text, including mention of the fact that some genes have been associated with both syndromic and non-syndromic forms of hearing loss.

2.The logical flow of the manuscript can be improved. In few points the authors seem to move forward on a different subject but then come back to the previous one (for example, in section 4 the mention to additional clinical symptoms -which are present in a minority of patients- may probably be moved after the description of the ear morphology in POU3F4 patients). I would also suggest verifying repetition of the same concepts in different sections.

Answer: As requested, the mention to additional clinical symptoms in section 4 was moved after the description of the inner ear morphology. The manuscript has been revised to avoid repetitions.

  1. Given that this is a review article, I would suggest making better use of the illustrations to convey the most important information also in a visual way. Currently, only two Figures are included, and they are not particularly clear or informative.

Figure 1 is difficult to interpret if you are not an expert, it lacks clear labelling and any reference to the size/dimension. Are all the three images at the same magnification? If possible, adding a drawing of the cochlear structure would help the interpretation of the radiological images. The legend to the figure could be improved. The order of the panels (A, B, C) should follow the same order in which the panels are mentioned in the text (currently, panel 1C is mentioned before panel 1B).

Answer: Thank you for these helpful comments. To better display the typical anatomical features of an IP-III cochlea in comparison with an healthy cochlea, a new set of panels have been selected for Figure 1. In the new Figure 1, panels A and B refer to a healthy cochlea, panels C and D to an IP-III cochlea. Panels A and C are CT scans, panels B and D are MRI images. Labelling has been improved and scale bars have been added. The figure legend has been revised. Also, the order of the references to the figure in the main text has been adjusted accordingly.

Also, Figure 2 can be implemented to convey more information. For instance, the authors may add a panel with the gene structure, and/or a panel with the consensus sequence logo of the Pou3f4 DNA binding site (publicly available in websites such as the JASPAR CORE database).

Answer: Thank you for these helpful comments. Figure 2 has been changed as suggested. Specifically, a schematic representation of POU3F4 gene with the only coding exon and a panel with the consensus binding sequence reported on the JASPAR database have been added.

4. Among possible targets of Pou3f4, the authors do not mention Otx2 (Inoue F, Kurokawa D, Takahashi M, Aizawa S. Gbx2 directly restricts Otx2 expression to forebrain and midbrain, competing with class III POU factors. Mol Cell Biol. 2012 Jul;32(13):2618-27. doi: 10.1128/MCB.00083-12), which has been demonstrated to be regulated by POU transcription factors, including Pou3f4, by binding a non-canonical target sequence.

Also, there is no mention to the fact that Pou3f4, as other POU transcription factors, can act as a monomer or a dimer (homodimer or heterodimers), thus making even more complicated to define precisely its regulatory network.

Answer: The authors thank the referee for these suggestions. A paragraph describing the role of POU III class transcription factors in the developing midbrain/hindbrain axes has been added to the text (page 4-5). The reference to the work by Inoue et al. has been added to the reference list. The existence of dimerization mechanisms in the POU family of transcription factors and its influence on sequence recognition has been mentioned in the main text (page 3).

  1. When presenting POU3F4 variants that have been reported in the literature (Table 1), it would be beneficial if the authors will evaluate the pathogenicity of the variants with some dedicated guidelines, for example using ACMG criteria, instead than reporting the pathogenicity assignment from the original work. In addition, it could be useful to report whether the mutations hit a specific functional domain/motif of the protein (e.g. POUs, POUh, NLS, linker). Finally, please specify that the table reports only SNVs and small indels, not structural variants (larger deletions/inversions, etc.).

Answer: As reported in the main text, in most of the cases the authors of the original publications assigned the pathogenicity according to the current ACMG guidelines. For clarity, we added a second table (Table 2) including the information extracted from the NCBI ClinVar and SNP databases, with the corresponding pathogenicity assignment and MAF, when available. Also, information on the domain affected by the individual variants has been added in the table and in the main text, as suggested. We have specified that table 1 only reports SNVs and small indels, not structural variants (larger deletions/inversions) and larger chromosomal rearrangements.

  1. Is it possible that some of the reported pathogenic variant will cause a gain of function of the transcription factor (for example altering the specificity of DNA binding or the specific interaction with protein partners)?

Answer: Although in principle possible, no such occurrence is known to the authors. Therefore, no modification of the manuscript was made based on this comment.

  1. The term “mental retardation” would be better substituted with “intellectual disability”, with an indication of the severity -if available.

Answer: The terminology has been modified as suggested. No precise information about the severity of intellectual disability is available to the authors.

Comments on the Quality of English Language

There are some grammar errors and typos throughout the manuscript. The manuscript would benefit from proof reading.

Answer: The manuscript has been proofread for typos and grammar.

Reviewer 4 Report

The review paper of POU3F4-related hearing loss is good.

Additional descriptions of expected treatment such as gene therapy or stem cell therapy for POU3F4-related hearing loss are required before conclusion. 

Author Response

Reviewer 4

The review paper of POU3F4-related hearing loss is good.

Additional descriptions of expected treatment such as gene therapy or stem cell therapy for POU3F4-related hearing loss are required before conclusion. 

Answer: The authors sincerely thank the reviewer of the manuscript for the positive comment and the helpful suggestion. A comment on the possible therapeutic options for POU3F4-related hearing loss has been added to the conclusions.

Round 2

Reviewer 1 Report

The authors have improved the manuscript with some of my recommendations. I have no further questions.

Author Response

The authors sincerely thank the reviewer for the suggestions provided to improve the manuscript.

Reviewer 3 Report

Overall, I appreciated the work the authors have done to suitably address most of the concerns I raised on the manuscript.

The introduction of Table 2, however, is quite redundant with Table 1 and does not really add much to the message of the paper. The authors may try to merge the two tables in just one, and avoid to list information that is not available for most variants (such as MAF). Instead, what I was previously suggesting -and still consider a key issue- was to provide themselves (asking the help of a medical/clinical geneticist if they need) a consistent classification of all previously reported variants using ACMG criteria (which of course might correspond to the ClinVar classification, where available). This will allow to assign to all variants, included those that are now classified as NA, a putative pathogenic status (P, LP, VUS, LB, etc.), which is what you will expect if those variants were to be reported to a patient after genetic screening. For instance, following the criteria indicated by Richards et al (doi: 10.1038/gim.2015.30), most loss-of-function (truncating/null, PVS1) variants that are rare (absent from population databases, PM2) can be classified as likely pathogenic (LP), unless they are located at the very 3’end of the gene and preserve the DNA binding domains. I believe this additional effort would provide much more relevant information to the readers than the current Tables.

Author Response

The authors are thankful for the further suggestions and clarifications provided by the reviewer. The two tables have been merged into one and the sparse information on the MAF and the corresponding SNP ID has been removed, as suggested. We have performed a thorough evaluation of the pathogenicity of each variant according to the criteria recommended by the ACMG for hearing loss (Oza et al, 2018; doi: 10.1002/humu.23630) and we have included this information in Table 1. The pathogenicity that we have assigned often overlapped with the pathogenicity given by the authors of the original studies.